# High-performance coherent optical modulators based on thin-film lithium niobate platform

Mengyue Xu [1], Mingbo He[1], Hongguang Zhang[2,3], Jian Jian[1], Ying Pan[1], Xiaoyue Liu [1], Lifeng Chen [1], Xiangyu Meng[1], Hui Chen[1], Zhaohui Li[1], Xi Xiao [2,3✉], Shaohua Yu[2,3], Siyuan Yu[1] & Xinlun Cai [1✉]

The coherent transmission technology using digital signal processing and advanced modulation formats, is bringing networks closer to the theoretical capacity limit of optical fibres, the Shannon limit. The in-phase/quadrature electro-optic modulator that encodes information on both the amplitude and the phase of light, is one of the underpinning devices for the coherent transmission technology. Ideally, such modulator should feature a low loss, low drive voltage, large bandwidth, low chirp and compact footprint. However, these requirements have been only met on separate occasions. Here, we demonstrate integrated thin-film lithium niobate in-phase/quadrature modulators that fulfil these requirements simultaneously. The presented devices exhibit greatly improved overall performance (half-wave voltage, bandwidth and optical loss) over traditional lithium niobate counterparts, and support modulation data rate up to 320 Gbit s$^{-1}$. Our devices pave new routes for future high-speed, energy-efficient, and cost-effective communication networks.

[1] State Key Laboratory of Optoelectronic Materials and Technologies and School of Electronics and Information Technology, Sun Yat-sen University, 510006 Guangzhou, China. [2] National Information Optoelectronics Innovation Center, China Information and Communication Technologies Group Corporation (CICT), 430074 Wuhan, China. [3] State Key Laboratory of Optical Communication Technologies and Networks, China Information and Communication Technologies Group Corporation (CICT), 430074 Wuhan, China. ✉email: xxiao@wri.com.cn; caixlun5@mail.sysu.edu.cn

Over decades and across all levels of optical networks, the global internet traffic has experienced continuous growth at an enormous rate[1,2]. To keep up with this ever-increasing demand, the digital coherent transmission technology has been introduced for long-haul communication links and is allowing networks to approach the maximum achievable capacity of optical fibres, known as the Shannon limit[3–7]. This technology is now expected to penetrate into the rapidly growing, capacity-hungry short reach links, such as metro and data-centre inter-connects, where an in-phase/quadrature (IQ) modulator must be operated in a small space, while featuring low loss, low drive voltages, and large bandwidths[8–10]. For nearly a decade, IQ modulators based on low-index-contrast lithium niobate (LiNbO$_3$, LN) waveguides have been the mainstay for generating the advanced modulation formats[11–14]. Although these mod-ulators have enjoyed tremendous success in long-haul coherent networks, their performance is already reaching a limit because the low-index-contrast LN waveguides cannot support them. To date, the off-the-shelf LN-based IQ modulators are still bulky and power-consuming with a moderate half-wave voltage ($V_\pi$) of 3.5 V requiring devices of at least 5 cm, and with little hope for further improving the electric–optic (EO) bandwidth (typically around 35 GHz), which limits their practical short reach applications.

Tremendous efforts have been made to realise small-footprint and high-performance IQ modulators in various material plat-forms, including silicon (Si), indium phosphide (InP), polymers, and plasmonics[15–32]. Although these platforms normally offer the advantages of compact footprints and large bandwidths, each type of modulator has its limitations, including large $V_\pi$ (Si), high optical loss (plasmonics), nonlinear response (InP), or doubts over long-term stability (polymer). The development of an ideal IQ modulator that possesses all the desired characteristics simultaneously remains a challenge, mainly due to limitations from underlying materials.

LN-on-insulator (LNOI) has recently emerged as an appealing material platform for compact and high-performance mod-ulators, on which high-contrast waveguides with strong optical confinement can be formed by simply etching the device layer of an LNOI wafer[33–46]. This approach unlocks new levels of per-formance and scales in LN modulators because it overcomes the fundamental voltage-bandwidth-size trade-off in conventional low-index-contrast LN modulators. Recently, LNOI-based Mach–Zehnder modulators (MZMs) with low drive voltages and ultrahigh EO bandwidths, which significantly outperform their conventional counterparts, have been demonstrated (see Supplementary Note 1).

Here, we demonstrate the IQ modulator based on the LNOI platform, which is capable of encoding signals with advanced modulation formats, such as quadrature phase-shift keying (QPSK) and quadrature amplitude modulation (QAM) signals. The pro-posed device features low optical loss, low $V_\pi$, ultrahigh EO bandwidth, and much smaller footprint than the conventional LN counterpart. Furthermore, QPSK modulation up to 220 Gbit s$^{-1}$ (110 Gbaud), and 16 QAM modulation up to 320 Gbit s$^{-1}$ (80 Gbaud), are successfully demonstrated.

## Results

**Device design.** Figure 1a illustrates the schematic of the proposed device. The LNOI-based IQ modulator is constructed by nesting two parallel travelling-wave MZMs as I and Q components, respectively. The optical power splitters/combiners are imple-mented by 1 × 2 multimode interference (MMI) couplers (see Supplementary Note 2, Supplementary Figs. 1 and 2). Both I and Q MZMs are balanced travelling-wave modulators with

ground–signal–ground (GSG) microelectrodes, where the two LN arms lie in the gaps of the ground and signal electrodes. (Fig. 1b) The device operates in a single-drive push–pull configuration, so that applied microwave fields induce phase shifts with an equal magnitude but opposite sign in both arms, leading to nearly chirp-free modulation. Two thermo-optic (TO) phase shifters (DC1 and DC2) are employed to control the modulation bias points of the I and Q branches independently. (Fig. 1c) For QPSK and QAM modulation, the bias points need to be set at null points. A third TO phase shifter (DC3) introduces a static π/2 phase shift between the modulated signals from the two sub-MZMs, which puts them in quadrature to each other. To couple light in and out of the device, amorphous silicon/LN hybrid grating couplers are used for transverse-electric (TE) mode coupling[47].

The cross-section of the LN waveguides, together with the travelling-wave electrodes, are optimised to achieve a low $V_\pi$ and a large EO bandwidth simultaneously. The LN waveguides in the phase modulation sections have a top width $w$ of 4 μm, a slab thickness $s$ of 300 nm and a rib height $h$ of 300 nm. The sidewall of the waveguide is tilted with an angle of 64° (Fig. 1d). The lithography and etching processes were optimised to achieve smooth sidewalls, which was confirmed by atomic force microscope (AFM) measurements (see Supplementary Note 3 and Supplementary Fig. 3). The propagation loss of single-mode and 4-um-wide LN waveguides were measured to be 0.3 dB cm$^{-1}$ and 0.15 dB cm$^{-1}$, respectively (see Supplementary Fig. 4). The gap between the LN waveguides and electrodes was set to 1.5 μm. These parameters are devised carefully to balance the trade-off between the voltage-length product ($V_\pi L$) and bandwidth-voltage ratio (BW/$V_\pi$), which are two key figures of merit of optical modulators (see Supplementary Note 4 and Supplementary Fig. 5). The coplanar travelling-wave electrodes are designed to simultaneously achieve broadband impedance matching, as well as velocity matching of the microwave and optical signals. The thickness of the electrodes $t$ was set to 900 nm; the gaps between the signal and ground electrodes were set to 7 μm, and the widths of the signal $w_s$ and ground $w_g$ electrodes were set to 19.5 and 80 μm, respectively (Fig. 1e).

**Device characteristics.** The fabricated LNOI IQ modulators with arm lengths of 7.5 mm and 13 mm are measured in detail. We first measured the direct current transmission with TO phase shifters. Figure 2a shows the TO transmission curve as a function of the applied voltage. The length of the TO phase shifter is only 160 μm with a resistance of 760 Ω, and the required voltages for biasing at quadrature and null are 3.55 and 7.2 V, corresponding to power dissipations of 16.6 and 68.2 mW, respectively. Instead of the EO effect, the TO effect is used here for DC bias voltage control, offering two distinct advantages. First, the TO effect is much stronger compared to the EO effect, allowing for much compact device size. In our case, the length of the EO phase shifter would have been more than 5 mm for the same amount of voltage. Second, LN is well-known for its drifts in the DC bias point upon the application of a static electric field, which is a phenomenon that originates from the piezoelectric nature of the material[11,48,49]. This drift must be compensated by a fast feedback loop for practical use. This phenomenon is absent in TO phase shifters, and a much simpler control scheme could be used instead. To provide a comparative study, we record the output power from an LNOI MZM biased at quadrature using both TO and EO phase shifters. Figure 2b shows the measured results in 30 min, confirming that the DC bias point with the TO effect is much more stable than with the EO effect. It should be noted that the TO phase shifter consumes static power, while EO phase

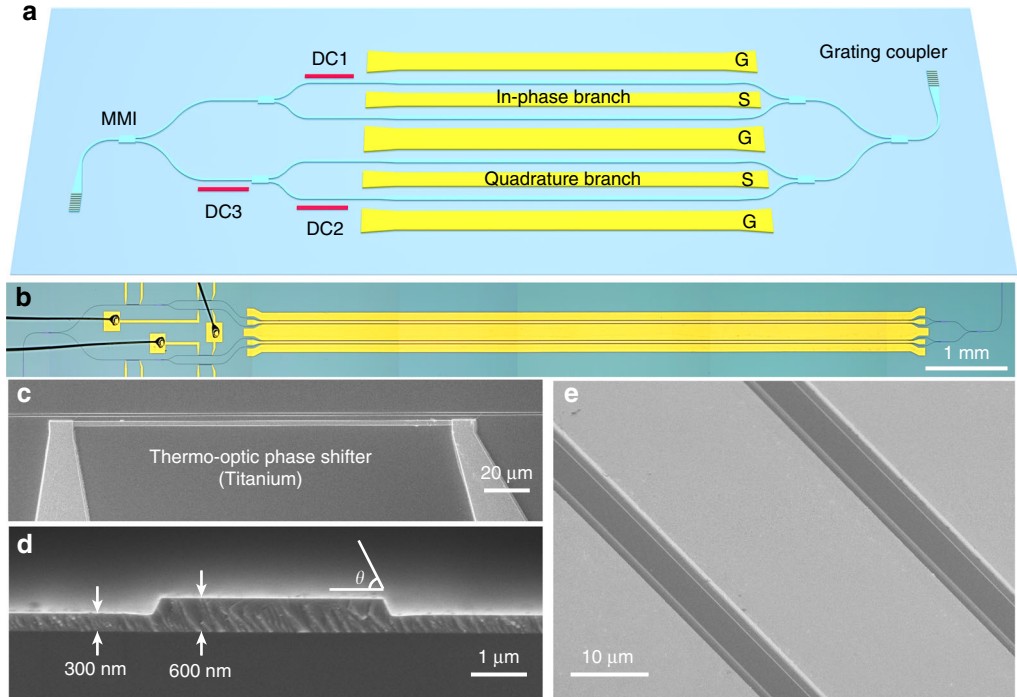

**Fig. 1 IQ modulator on the LNOI platform. a** Schematic of an LNOI-based IQ modulator. **b** Microscope image of the fabricated chip. **c** Scanning electron microscopy (SEM) image of the thermos-optic phase shifter. **d** SEM image of the cross-section of the LN waveguides. **e** SEM image the gold electrodes and the LN waveguides.

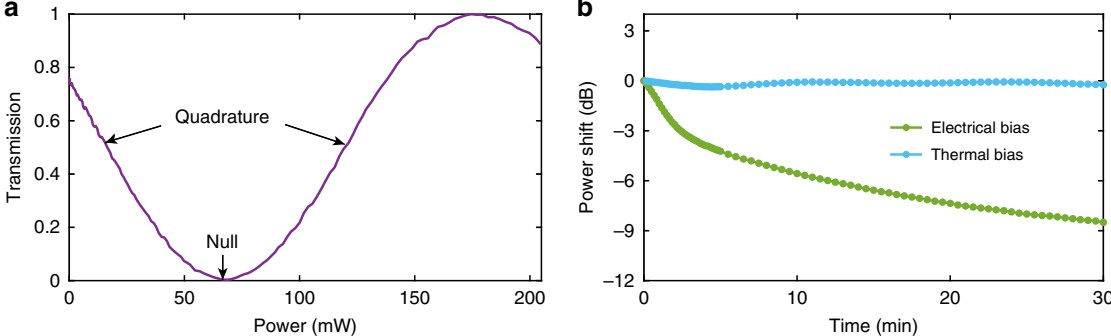

**Fig. 2 TO phase shifters performance. a** TO transmission curve as a function of the power dissipations. **b** Power shift from an MZM biased at quadrature using EO (green line) and TO (blue line) phase shifters as a function of the operating time.

shifter does not. We also fabricated an IQ modulator with EO phase shifters for further comparison (Supplementary Note 5, Supplementary Fig. 6 and Supplementary Table 1).

Next, we measured the $V_\pi$ for both devices with 100 kHz triangular voltage sweeps. The measured $V_\pi$ for the 7.5 and 13 mm devices are 3.1 and 1.9 V, corresponding to $V_\pi L$ of 2.3 and 2.4 V cm, respectively (Fig. 3a, b). The inset of Fig. 3a shows the transmission of one of the sub-MZMs on a logarithmic scale, indicating a measured extinction ratio of >25 dB. We have fabricated more than ten devices on the same chip, and the measured extinction ratios are between 24 and 28 dB. Figure 3b shows the optical transmission at different wavelengths, showing broadband operation in the whole C-band.

We then characterised the small-signal EO bandwidth ($S_{21}$ parameter) and electrical reflections ($S_{11}$) of the fabricated devices. For the 13-mm device, the measured 3-dB EO bandwidths of both I and Q MZMs are greater than 48 GHz with a reference frequency of 1.5 GHz. For the 7.5-mm device, the EO bandwidth is greater than 67 GHz (Fig. 4a), which is beyond the measurement limit of our vector network analyser (VNA). These voltage and bandwidth parameters indicate a much better performance over conventional LN modulators and are well suited for high-speed operation beyond 100 Gbaud. The input return losses ($S_{11}$ parameter) of both devices are less than −18 dB at up to 67 GHz, which are small enough for practical use (Fig. 4b).

The fibre-to-fibre insertion losses at peak transmission are measured to be 8.6 dB and 8.25 dB for 13 mm and 7.5 mm device, respectively. The coupling loss of the grating couplers is 3.4 dB/facet. Therefore, the on-chip losses are 1.8 dB and 1.45 dB for 13 mm and 7.5 mm device, respectively. The coupling loss can be further improved by replacing grating coupler with edge-coupled spot-size converters, with fibre-to-fibre losses below 4 dB practically achievable.

**Data modulation.** We use our low-$V_\pi$, large-bandwidth, and low-loss 13-mm LNOI IQ modulators to generate advanced modulation formats of up to 320 Gb s$^{-1}$. The experimental setup is

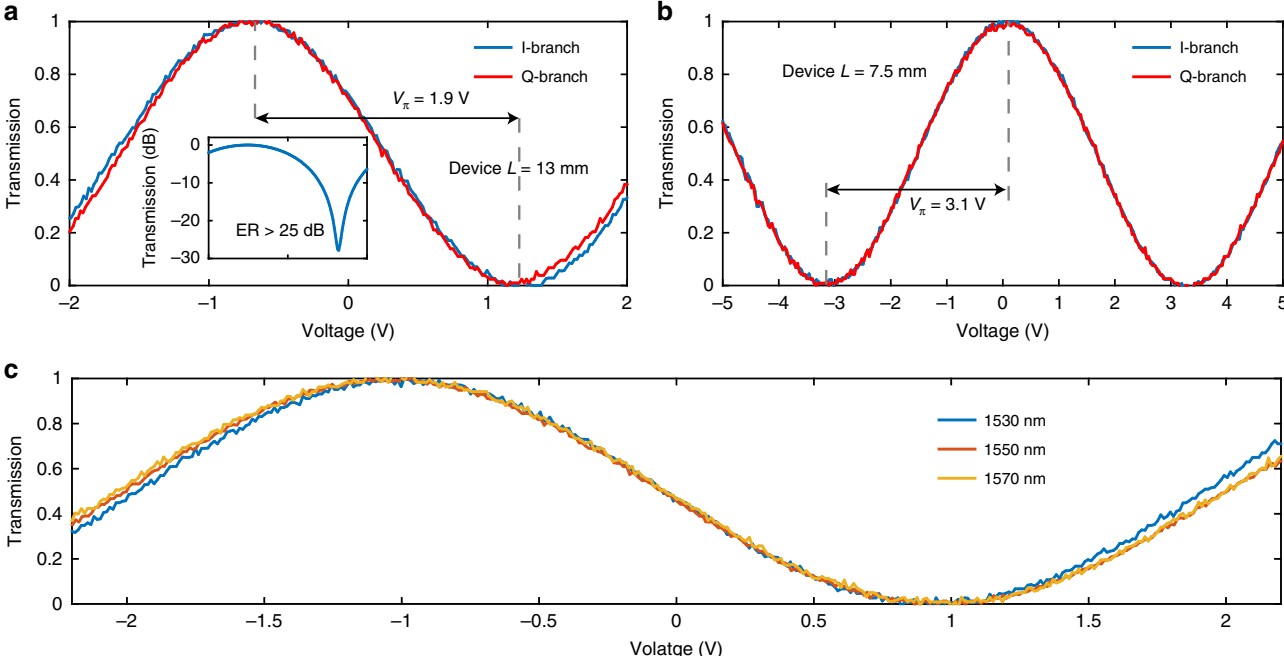

**Fig. 3 Static EO characteristics. a, b** Normalised optical transmission of both branches of the 13-mm and 7.5-mm devices as a function of the applied voltage, showing $V_\pi$ of 1.9 V and 3.1 V, respectively. The inset of **a** shows the measured normalised transmission on a logarithmic scale, showing an extinction ratio greater than 25 dB. **c** Measured optical transmission at different wavelengths.

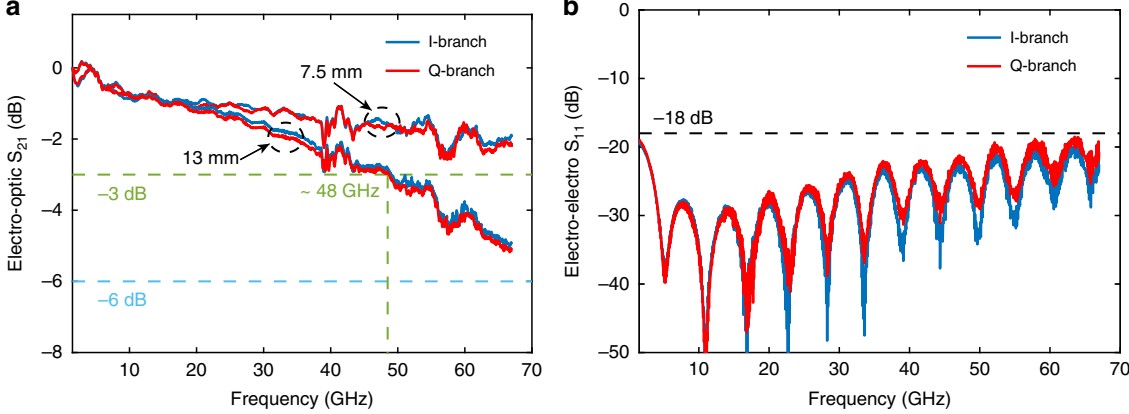

**Fig. 4 Small-signal response. a** EO bandwidths ($S_{21}$ parameter) and **b** electrical reflection $S_{11}$ of the 13-mm IQ modulator.

shown in Fig. 5a. First, we demonstrate the generation of QPSK, one of the most widely deployed formats in coherent transmission systems, for performance benchmarking. Figure 5b–e summarises the measured constellation diagrams at 60, 80, 100, and 110 Gbaud, corresponding to data-transmission rates of 120, 160, 200, and 220 Gb s$^{-1}$, respectively. All the modulated QPSK signals yield very good bit error rate (BER) performance well below the KP4 forward error correction (FEC) limit of $2 \times 10^{-4}$. Moreover, error-free operations at 80 and 100 Gbaud were achieved with BER $< 1 \times 10^{-9}$. We then use 16 QAM modulation formats at high baud rate to further increase the data rates and investigate the signal-to-noise ratio (SNR), a common measure for the fidelity of modulated signals. The constellation diagrams at 60 and 80 Gbaud with QAM are shown in Fig. 5f, g. With the 60 Gbaud 16 QAM (data rate of 240 Gbit s$^{-1}$), we can achieve a low BER of $8.6 \times 10^{-5}$. With the 80 Gbaud 16 QAM (data rate of 320 Gbit s$^{-1}$), the measured BER of $8.4 \times 10^{-3}$ is still within the tolerance of the soft-decision forward error correction (SD-FEC) limit of $4 \times 10^{-2}$. The back-to-back (B2B) BER curves at

60 Gbaud 16 QAM, shown in Fig. 5h. The BERs are also well below the SD-FEC limit and no error floor can be observed in the measurement. The high SNR demonstrated here benefits from a combination of linear EO response, low insertion loss and pure phase modulation, all of which are derived from the Pockels effect in the LN material.

## Discussion

As presented above, the reported IQ modulator demonstrates ultrahigh speed and high-signal fidelity when encoding information into the amplitude and phase of the light. As far as we are aware, this is the first reported successful demonstration of IQ modulators based on LNOI platforms. The overall performance of optical modulators in terms of voltage, bandwidth, and footprint can be evaluated by two figures of merit: the $V_\pi L$ and BW/$V_\pi$. The present device exhibits $V_\pi L$ and BW/$V_\pi$ of around 2.5 V cm and 25 GHz V$^{-1}$, significantly outperforming conventional LN counterparts. The overall length of the present device is ~15 mm, which is small enough to fit into compact coherent

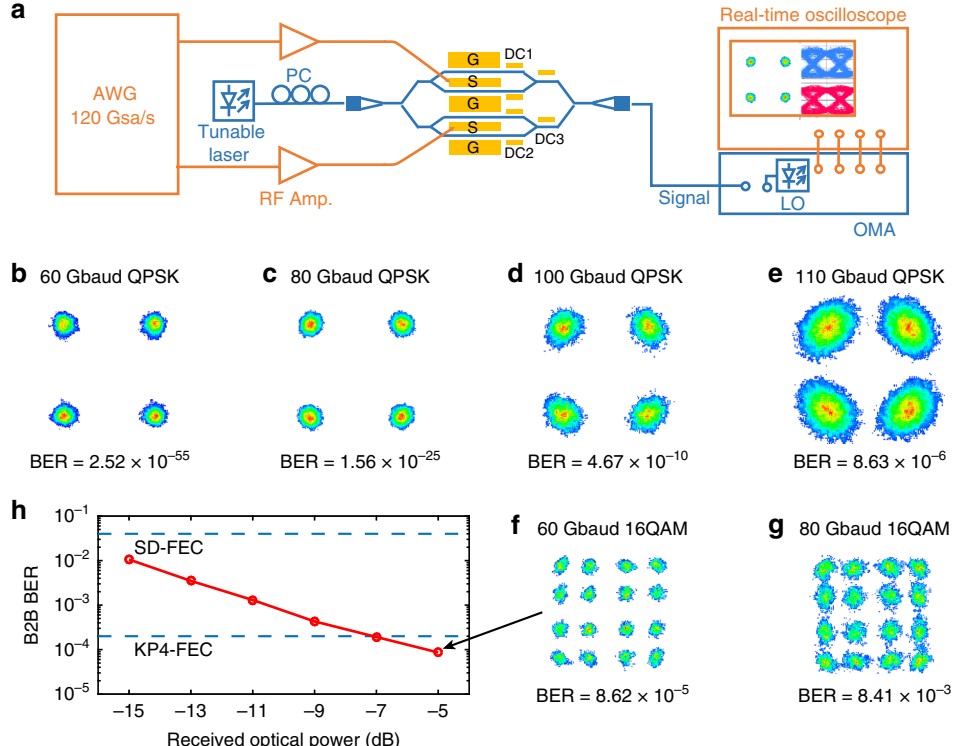

**Fig. 5 Data modulation testing. a** Experimental setup for coherent data transmission. AWG: arbitrary waveform generator, LO: local oscillator, OMA: optical modulation analyser, PC: polarisation controller. **b**–**g** Constellation diagram for QPSK signals with symbol rates of 60, 80, 100, and 110 Gbaud and 16 QAM signals with symbol rates of 60 and 80 Gbaud. **h** Measured curve of BER versus the received optical power for 60 Gbaud 16 QAM signal.

**Table 1 Comparison of several performance metrics of IQ modulators.**

|  | $V_\pi$ (V) | 3-dB EO bandwidth (GHz) | Length of modulation area (mm) | On-chip loss (dB) | Data rate (Gb/s) (BER) |
|---|---|---|---|---|---|
| SOI[19] | 7.5 | 32 | 4.5 | 6.8 | 360 ($2 \times 10^{-2}$) |
| InP[16,17] | 1.5 | 67 | ~3.6 | ~2 | 448 ($6.68 \times 10^{-3}$) |
| InP[18] | 1.7 | 43 | 4 | 7[a] | 64 |
| GaAs[24] | 3 | 27 | 30 | <8 | 150 ($1.1 \times 10^{-2}$) |
| SOH[28] | 1.6 | NA | 0.6 | 8.5[b] | 400 ($1.7 \times 10^{-2}$) |
| Plasmonic[32] | 8.67 | >500 | 0.015[c] | 11.2 | 400 ($4.5 \times 10^{-2}$) |
| Commercial LN[51] | 3.5 | 35 | 30~80 | NA | 256 |
| This work | 1.9 | ~48 | 13 | 1.8 | 320 ($8.41 \times 10^{-3}$) |
|  | 3.1 | >67 | 7.5 | 1.45 |  |

NA: not available.
[a]This value was calculated from the 8 dB insertion loss and 1 dB loss in the spot-size converter.
[b]This value was calculated from the 17.5 dB fibre-to-fibre loss and 9 dB off-chip coupling loss.
[c]This value was calculated from the reported $V_\pi L$ of 130 V µm and length of 15 µm.

transceiver packages, such as the CFP2-ACO (C-form factor pluggable analogue coherent optics) and QSFP-DD (quad small form factor pluggable double density).

In Table 1, we compare the performance of the present device with the state-of-the-art, including commercial LN IQ modulators, and IQ modulators based on silicon, InP, GaAs, silicon–organic-hybrid (SOH) and plasmonics. Conventional LN IQ modulators are also included as a benchmark. Clearly, the present device features the best optical loss compared among the others, and to the best of our knowledge, this is the lowest insertion loss ever achieved in IQ modulators. This characteristic makes the present device suitable for applications in short reach links, where the optical power budget becomes critical owing to prohibitive deployment of optical amplifiers. The $V_\pi$ and bandwidth of the present device are also very appealing for high-speed and low-power consumption operations, which are comparable to those of

the best InP modulators. Furthermore, in contrast to InP-based IQ modulators, in which thermoelectric cooling (TEC) is indispensable for reliable operation, the LNOI-based IQ modulator could be operated without TEC, which is highly desirable for low-cost applications. We believe that the EO bandwidth of the present device could be further extended beyond 100 GHz without compromising the half-wave voltage, by further optimising the travelling-wave electrode (see Supplementary Note 4). This could support data rates of over 200 Gbaud. By further integrating on-chip polarisation combiners, a single LNOI-based modulator could operate at a data rate of >1 Tb s$^{-1}$ using, for example, 16 QAM modulation at 200 Gbaud. The demonstrated LNOI platform could therefore lead to a paradigm shift in building compact and high-performance IQ modulators, offering a crucial edge for future ultra-fast and low-power consumption optical fibre interconnects.

## Methods

**Device fabrication.** The devices were fabricated on a commercial X-cut LNOI wafer from NANOLN. We used electron-beam lithography (EBL) to define waveguide patterns after spinning Hydrogen silsesquioxane (HSQ). The 300-nm-high ridges of the LN waveguides were formed in an optimised argon plasma in an inductively coupled plasma etching system. Afterwards, we deposited a 220-nm amorphous-Si (a-Si) layer on the patterned LN waveguides and spin-coated HSQ on the a-Si for EBL. The Si/LN grating couplers were used for off-chip coupling. Then, we used a polymethyl methacrylate resist for a lift-off process to produce the 200-nm-thick TO phase shifters. Finally, 900-nm-thick gold travelling-wave electrodes were patterned through a lift-off process.

**High-speed data modulation.** At the transmitter, the two independent RF signals with a length of a pseudo-random bit sequence of $2^{15}-1$ were generated from a 120-GSa/s arbitrary waveform generator (Keysight, M8194A). A root-raised-cosine filter was used for pulse shaping in the frequency domain. After being amplified by linear amplifiers (SHF S807C, 3 dB BW: 55 GHz), the output signals were fed into the IQ modulator by an RF probe (ground–signal-ground–signal-ground (GSGSG) configuration, 3 dB BW > 67 GHz). A second GSGSG RF probe was used to terminate the end of the transmission line with a 50-Ω load to avoid back-reflected signals.

Light from a tunable laser (Santec TSL-550) with 17 dBm output power was coupled into and collected from the chip via amorphous-Si/LN grating couplers. A polarisation controller was used to ensure transverse-electric mode input. An optical modulation analyser (OMA, Keysight N4391B) with a four-channel real-time oscilloscope (Keysight UXR0704A, 256 GSa/s) served as a receiver. The modulated signal was received B2B and mixed with an internal local oscillator in the OMA. A series of digital signal processing steps, including low-pass filtering, polarisation demultiplexing, and feed-forward compensation, were included in the vector signal analysis (Keysight) software. The data decoded from the OMA were used for BER and error vector magnitude (EVM) measurements. The EVM describes the deviation of a measured symbol point from its ideal position in the constellation diagram and can be used to reliably estimate the BER values assuming the signal is distorted by additive white Gaussian noise only[50].

**Modulator energy considerations.** For 16 QAM modulation, we can estimate the energy consumption per bit dissipated in the travelling-wave IQ modulator as $W_{bit,16QAM} = 2 \times V_{rms}^2/(BR)$, where $V_{rms}$ is the root-mean-square voltage of the electrical PAM4 signal, $B$ is the total bit rate and $R$ is the equivalent resistor of 50 Ω. The value of $V_{rms}$ applied to the one MZM was measured directly using the oscilloscope. For 60 Gbaud 16 QAM ($B = 240$ Gbit s$^{-1}$) modulation experiment with $V_{rms} = 0.85$ V, we can calculate an energy consumption of 120 fJ/bit. For 80 Gbaud 16 QAM ($B = 320$ Gbit s$^{-1}$) modulation experiment with $V_{rms} = 0.7$ V, we find a lower power consumption of 61 fJ/bit. To further reduce the energy consumption, we fabricated a LN MZM with a record low $V_{\pi}$ of 1.25 V while maintaining a high modulation bandwidth (see Supplementary Note 6 and Supplementary Fig. 7).

## Data availability

All the data supporting the findings in this study are available in the paper and Supplementary Information. Additional data related to this paper are available from the corresponding authors upon request.

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

## Acknowledgements

This work was supported by the National Key R&D Program of China (Grant Nos. 2019YFA0705000, 2019YFB1803900), the National Natural Science Foundation of China (Grant Nos. 11690031, 11761131001), the Key R&D Program of Guangdong Province (Grant No. 2018B030329001), the Local Innovative and Research Teams Project of Guangdong Pearl River Talents Program (Grant No. 2017BT01X121), the Innovation Fund of WNLO (Grant No. 2018WNLOKF010) and the Project of Key Laboratory of Radar Imaging and Microwave Photonics, Ministry of Education (Grant No. RIMP2019003).

## Author contributions

X.C., M.X., and X.X. conceived device design. M.H., J.J., and P.Y. carried out the LN fabrication. M.X., H.Z., and Z.L. carried out the measurement. S.H.Y., S.Y.Y., X.X., and X.C. carried out the data analysis. All authors contributed to the writing. X.C. finalized the paper. S.Y.Y., S.H.Y., X.X., and X.C. supervised the project.

## Competing interests

H.Z., X.X., and S.H.Y. are involved in developing silicon photonics technologies at China Information and Communication Technologies Group Corporation. The remaining authors declare no competing interests.
