## [Peer Review File · Nature Communications]

Reviewers' comments:

Reviewer #1 (Remarks to the Author):

This is a paper by the same authors who published a paper in nature photonics on a very similar device ("High-performance hybrid silicon and lithium niobate Mach-Zehnder modulators for 100 Gbit s⁻¹ and beyond"). The improvement is that they only use the larger multi-mode LN waveguide instead of using evanescent coupled small LN waveguides, and the device shows a very low insertion loss. At the same time, the authors use coherent equipment to do the coherent modulation test at a higher baud rate (maximum 110 GBaud). The results are pretty good. There are some technical questions about the manuscript, which follow below.

What are the modal properties of the LN waveguide? It looks like a multimode waveguide. Why didn't the authors choose a single mode waveguide? Only to have a lower optical loss? What is the loss of the MMI? What do they give up by going multimode?

The caption of Fig.2 is wrong. The EO line is green and the TO line is blue.

The V_{pi} description in Fig.3 caption is wrong. It should be 3.1 V instead of 7.5 mm. The wavelength data is (c) not (b).

The intrinsic extinction ratio is around 25 dB. Is this a general phenomenon or a singular case? What is the limitation for this performance?

Regarding the s₂₁ response, there are two large ripples at around 40 GHz and 56 GHz for both the 7.5-mm-long and 13-mm-long devices. What is the reason for this?

In fig.5, it is mysterious that constellation diagram at 80 GBaud is better than that at 60 GBaud, but the BER is worse? The constellation diagram is swapped or something else affects this?

For the 16-QAM measurement, the device is over driven from the constellation diagram. Lower driving voltage is suggested to exploit the performance of the device, especially for Fig.5(h).

Reviewer #2 (Remarks to the Author):

This paper presents what is claimed to be the first demonstration of an IQ modulator in thin film lithium niobate. The structure is traditional, consisting of two parallel Mach-Zehnder interferometers nested within a further Mach-Zehnder structure with a 90 degree phase shift introduced before combination.

The Mach-Zehnder structures are very traditional similar to [1] with velocity and impedance matched electrodes in a push-pull configuration. The device length in this submission are 7.5mm and 13mm. The Loncar demonstration [1] has longer electrodes (10mm and 20mm), but with similar geometry. Loncar shows a V_{pi} of 1.4V with a bandwidth over 40GHz, while this submission demonstrates 1.9V with a bandwidth over 60GHz. So the drive voltage is not the highest reported.

The bandwidth is also not the highest reported. The publication [2] by Prather shows bandwidths can exceed 100GHz (approaching 1 THz) - however with a degraded V_{pi} of 3.8V.

The submitted paper quotes an optical insertion loss of 0.5dB/cm, while [1] quotes 0.5dB for the entire structure. The submitted paper does not report on the total insertion loss of the chip and in particular does not report the insertion loss due to propagation in proximity of the modulation or bias electrodes - it is important that these numbers be quoted as it is important that the electrodes do not interact strongly with the light for practical operation.

The submitted manuscript devotes a significant portion of the paper to reporting on the thermo-optic phase controllers. After a quick scan of the literature, I did not find any other report of thermo-optic switches on LNOI, so this may be innovative. However, the analysis done by the authors is not convincing. While electro-optic behaviour can be assessed based on the voltage required, the thermo-optic switches cannot be

compared on voltage alone. The current must also be considered in order to assess the power draw.

I also do not agree with the authors about the 'LN is well-known for its drifts in the DC bias point upon the application of a static electric field, which is a phenomenon that originates from the piezoelectric nature of the material'. The authors must provide a reference for this statement. Bias control is necessary for interferometric lithium niobate devices due to the pyroelectric effect (changes in temperature cause a change in surface charge which in turn changes the refractive index). I know of many lithium niobate devices that do utilise an electro-optic bias and are able to maintain stability. Perhaps the authors are effectively controlling the temperature of the LNOI film with their TO heaters? I do not believe that their analysis of the instability of their own device when driven electro-optically provides any scientific insight and hence, I would encourage presenting the thermo-optic switches in a more technical journal where it can be the focus and referencing this in a higher profile paper describing the 'world first' IQ measurement.

Overall, the LNOI technology presented here does not, in my view, represent any significant advancement in the state of the art; however, the implementation of a IQ modulator may indeed be a world first, and perhaps this is significant enough to warrant publication in a high profile journal such as Nature Communications. However, in order to achieve this level, I think the focus needs to be placed more on the system demonstration that harnesses the IQ modulator (rather than the technology used to realise the chip itself) and it would need to be demonstrated that a record breaking transmission characteristic was achieved (such as modulation density, or power efficiency?).

I also note a number of typographical errors throughout the manuscript (for example "both branches of the 13-mm and 7.5-mm devices as a function of the applied voltage, showing V_n of 1.9 and 7.5 mm, respectively." - the last number should be the V_{pi} of the 7.5mm electrode, not its length). There are several others.

[1] Wang, C., Zhang, M., Chen, X. et al. Integrated lithium niobate electro-optic modulators operating at CMOS-compatible voltages. Nature 562, 101–104 (2018)

[2] Mercante AJ, Shi S, Yao P, Xie L, Weikle RM, Prather DW. Thin film lithium niobate electro-optic modulator with terahertz operating bandwidth. Optics express. 2018 May 28;26(11):14810-6.

Reviewer #3 (Remarks to the Author):

Lithium niobate is one of the most important materials for optical modulators. With the LNOI (lithium niobate on insulator) platform, new types of optical modulators were introduced in recent years. This kind of modulators has the advantages of small volume and high performance, such as low driving power, high bandwidth, etc. However, LNOI based modulators still face several important questions. 1. Can this modulator be used on short reach links, such as metro and data-center interconnects? Modulators for short reach links account for the most amount of the modulator industry production. In such application, the basic requirements for the modulator are small volume, low optical loss, low power consumption, and low cost. Currently the standard choice is the semiconductor (InP, for example) based modulators. InP modulator has the drawbacks of high optical loss and non-linearity. LNOI waveguide showed an ultra-low loss down to 0.03 dB/cm (Optica 4, 1536, 2017), and conventional LN modulator showed a good linearity. Therefore, LNOI modulator has a potential to be a strong competitor with InP modulators. 2. For LNOI modulator, which fabrication routine will be adopted? In other words, what is the future technology roadmap? Two complementary fabrication routines were used to make LNOI based modulators: monolithic and hybrid. Monolithic means direct etching of LN to form the waveguides (Nature 562, 101, 2018). This kind of modulator theoretically has a better device performance. An example of hybrid is SOI-LNOI modulator (Nat. Photon. 13, 359, 2019). This modulator has the advantage of the mature fabrication technology, and easiness to integrate with photo detectors (such as epitaxy of SiGe detector). Which fabrication routine will be the mainstream? This question is still not clear to the academy and the industry. The people in industry especially care about this question, because they concern the R&D inputs. 3. For conventional LN modulator,

a major problem is DC drift. This phenomenon is attributed to the piezoelectric or pyroelectric properties of the LN material. People spent a lot of efforts to solve this problem. For LNOI modulator, because it is a new research field, there is very few data or public report on the DC drift problem.

This manuscript reported the first monolithic IQ (in-phase/quadrature) LNOI modulator used for short reach links, solved the DC drift problem using TO (thermal-optic) phase shifters. The device dimension is small (15 μm), and the performance is excellent, such as low loss (1.45 dB), low $V_{\pi L}$ (2.5 V cm), high data rate (320 Gbit s⁻¹), and large bandwidth (> 67 GHz).

The drawback of this work is that TO phase shifters needed electric power to operate, which increased the power consumption.

This work provided the strong evidences to answer the three questions mentioned above. It will be very interesting to the people working on the modulators.

A publication of the manuscript is highly recommended. However, the authors should response the following comments before the publication.

1. The authors can give some comments on the comparison on the modulators fabricated by hybrid or monolithic. Since the authors had the experience and insight on SOI-LNOI hybrid modulator (Nat. Photon. 13, 359, 2019), and this manuscript was about monolithic modulator, the authors' comments would be very useful to the readers.
2. The authors can give some comments or data on the multimode interference couplers (MMIs) in this work. There is few report about MMI coupler on LNOI. Any information about it will be useful.
3. The authors used TO phase shifter to suppress the DC drift, but the TO effect would increase the power consumption of the modulator. Can EO effect be used for DC drift control in LNOI modulator?
4. The authors have to check the content of the manuscript carefully to avoid any miswritting, for example, in line 73, "Fig. 2b" should be "Fig. 1b".

The following response is for the referee reports for manuscript “High-performance Coherent Optical Modulators based on Thin-film Lithium Niobate Platform”. Blue text signifies the reviewers’ comments. Black text is our response. Black italic text is quotes from our revisions (and all figures included are also from our revisions).

Reviewer #1 (Remarks to the Author):

This is a paper by the same authors who published a paper in nature photonics on a very similar device (“High-performance hybrid silicon and lithium niobate Mach–Zehnder modulators for 100 Gbit s⁻¹ and beyond”). The improvement is that they only use the larger multi-mode LN waveguide instead of using evanescent coupled small LN waveguides, and the device shows a very low insertion loss. At the same time, the authors use coherent equipment to do the coherent modulation test at a higher baud rate (maximum 110 GBaud). The results are pretty good. There are some technical questions about the manuscript, which follow below.

We are grateful to the Referee for their time and effort in reviewing our work.

What are the modal properties of the LN waveguide? It looks like a multimode waveguide. Why didn't the authors choose a single mode waveguide? Only to have a lower optical loss?

Response: Thank you for your questions. In the phase modulation region, we adopt 4- μm -wide LN waveguide and this is indeed a multimode waveguide.

We choose 4- μm -wide LN waveguide not only to achieve low optical loss, but also to balance the trade-off between voltage-length product ($V_{\pi}L$) and modulation bandwidth. In our simulation, we find that a single mode waveguide indeed support lower value of $V_{\pi}L$, however, it will also lead to higher value of RF loss, which will degrade the modulation bandwidth. According to our simulation, a single mode waveguide is not an optimal choice for the overall performance of the modulator. We elaborate our design for the width of LN waveguide in phase modulation section in **Supplementary Note 4| Design of phase modulation waveguides and travelling-wave electrodes**. As depicted in **Supplementary Fig. 5**, we achieve the largest BW/ V_{π} value at the width of the LN waveguide $w=4 \mu\text{m}$ and the optimized electrodes gap is $7 \mu\text{m}$, and we adopt this design in our experiment.

Moreover, we design our device in such a way that only the fundamental mode is excited. We adopt single-mode waveguides for all the other parts of the device (including the access waveguides for MMI and all the bend waveguides) except the phase modulation parts. We use adiabatic waveguide tapers at both ends of the phase modulation parts to make sure that the no high order modes can be excited when the light propagates in the device. To better convey this information, we have added one section in the revised Supplementary Information for the detailed simulation results of the mode transition from the single mode waveguide to 4- μm -wide LN waveguide.

In the **Supplementary Note 2**: *“In the practical devices, we adopt single-mode waveguides (width of 1 μm) for all the other parts of the device (including the access waveguides for MMI and all the bend waveguides) except the phase modulation parts. We use adiabatic waveguide tapers to connect 1- μm -wide single-mode waveguides and 4- μm -wide phase modulation waveguides, which does not incur any conversion from fundamental mode to high-order modes (Supplementary Figure 2a). In this way, we can make sure that only the fundamental mode is excited when the light propagates in the device (Supplementary Figure 2b). In Fig. S2c, the simulated result shows*

the designed adiabatic taper can achieve a transmission of approximately 99.9 %.”

Supplementary Figure 2 | Mode transition. *a* Schematic of an adiabatic waveguide taper and mode profile of a 1- μm -wide waveguide and a 4- μm -wide waveguide, respectively. *b* SEM image of a MMI and adiabatic tapers. *c* Simulated transmission of the adiabatic waveguide taper as a function of taper length.”

What do they give up by going multimode?

Response: Thank you very much and this is an important question. We give up the device length for better modulation bandwidth, when we choose multimode waveguide instead of single mode. As mentioned above, the multimode waveguide for phase modulation is associated with larger value of $V_{\pi}L$, which is a figure of merit for device length and drive voltage. The design of LN travelling wave modulator is to achieve a balance among drive voltage, modulation bandwidth and device length. In our design, we choose to optimize the value of BW/V_{π} , which is a figure of merit for the modulation bandwidth and drive voltage, instead of $V_{\pi}L$. Here we put more priority to modulation bandwidth compare to device length. For example, the $V_{\pi}L$ for 1- μm -wide waveguide and 4- μm -wide waveguide are 2.5 Vcm and 2.65 Vcm, respectively. However, the BW/V_{π} for 1- μm -wide waveguide and 4- μm -wide are 17.2 GHz/V and 23.04 GHz/V.

What is the loss of the MMI?

Response: The measured excess loss of a LN MMI is ~ 0.054 dB. We added the following texts and figures in the revised Supplementary Information for the detailed design and measurement results for the MMI.

In the **Supplementary Note 2:** We employ 1×2 multimode interference (MMI) couplers to provide symmetric power splitting and combing in our modulators. The MMI couplers considered here are based on the etched LN waveguides with a slab thickness of 300 nm and a rib height of 300 nm. The length of the multimode region L_M was designed to be 64 μm , and the width of the multimode region W_M and the wider ends were set as 9.654 μm and 4.54 μm , respectively (Supplementary Figure 1a). Three identical 50- μm -long linear tapers are used as adiabatic tapers and connected to single-mode waveguides. Supplementary Figure 1b shows the simulated excess loss for different wavelengths.

To measure the excess loss of MMI coupler, we fabricated the MMI couplers arranged in a cascaded configuration, as depicted in Supplementary Figure 1c. Supplementary Figure 1d shows

the measured transmission of the MMI couplers at the wavelength of 1550 nm. We extract the excess loss of ~ 0.054 dB from the slope of the fitting line. Next, we experimentally demonstrated the performance of our 3-dB MMI couplers by integrating two MMI couplers into an unbalanced MZI (Supplementary Figure 1e). Supplementary Figure 1f shows the static extinction ratio is greater than 40 dB, which indicates that the power splitting ratio of the MMI is very close to 50:50.

In the practical devices, we adopt single-mode waveguides (width of $1 \mu\text{m}$) for all the other parts of the device (including the access waveguides for MMI and all the bend waveguides) except the phase modulation parts. We use adiabatic waveguide tapers to connect $1\text{-}\mu\text{m}$ -wide single-mode waveguides and $4\text{-}\mu\text{m}$ -wide phase modulation waveguides, which does not incur any conversion from fundamental mode to high-order modes (Supplementary Figure 2a). In this way, we can make sure that only the fundamental mode is excited when the light propagates in the device (Supplementary Figure 2b). In Fig. S2c, the simulated result shows the designed adiabatic taper can achieve a transmission of approximately 99.9 %.

Supplementary Figure 1 | MMI performance. *a* Schematic of an MMI coupler. *b* Simulated excess loss of the MMI for different wavelengths. *c* Schematic of the MMI couplers arranged in a cascaded configuration for loss measurement. *d* Transmittance measured at the output of the cascaded MMI couplers as a function of the number of branches (blue circle) a linear fit (red solid line). *e* Schematic of an unbalanced MZI with two 1×2 MMI couplers. *f* Measured static extinction ratio of the unbalanced MZI.

The caption of Fig.2 is wrong. The EO line is green and the TO line is blue.

The Vpi description in Fig.3 caption is wrong. It should be 3.1 V instead of 7.5 mm. The wavelength data is (c) not (b).

Response: Thank you very much indeed, and sorry for the careless mistakes. We have corrected the caption of Fig.2 and Fig. 3 in the revised manuscript.

The intrinsic extinction ratio is around 25 dB. Is this a general phenomenon or a singular case? What is the limitation for this performance?

Response: We have fabricated more than 10 devices on the same chip, and all of the measured extinction ratios are 24~28 dB. Therefore, we think this result is quite general. The 25 dB extinction ratio we report in the main text is the value for the device which we used for the data transmission experiment.

The value of extinction ratio is limited by two factors: the power-splitting ratio of MMI and the loss imbalance between two arms of MZM. We have designed and fabricated another batch of device to measure the power splitting ratio of MMI, and the results confirmed that the power splitting ratio of the MMI and it is very close to 50:50 (**Supplementary Note 2**). We believe the extinction ratio is currently limited by the unbalanced loss from two MZM arms. The lengths of the MZM arms are longer than 7.5 mm and 13 mm. Waveguides with lengths of this level are associated with unavoidable loss variations, resulting from fabrication imperfections like surface roughness and dimension fluctuations.

To clarify this, we add one sentence in the revised manuscript:

“We have fabricated more than ten devices on the same chip, and the measured extinction ratios are between 24 to 28 dB.”

Regarding the S_{21} response, there are two large ripples at around 40 GHz and 56 GHz for both the 7.5-mm-long and 13-mm-long devices. What is the reason for this?

Response: We believe the ripples around 40 GHz and 56 GHz are from the high-speed photodetector (Finisar 70 GHz XPDV3120), which we used for S_{21} measurement. We can always observe ripples around these two frequencies when we measured with this photodetector. The following figure shows the frequency response of at commercial modulator (purple line), which was measured with Finisar 70 GHz XPDV3120. The ripples appear at the frequency 38.5-44 GHz and 56-60 GHz (yellow regions).

In fig.5, it is mysterious that constellation diagram at 80 GBaud is better than that at 60 Gbaud, but the BER is worse? The constellation diagram is swapped or something else affects this?

Response: Thank you for your question. In the original manuscript, two different types of filters in the digital signal processing of optical modulation analyser (OMA) were used for 60 Gbaud (low-pass filtering) and 80 Gbaud (root raised cosine, RRC filtering) QPSK signals. The usage of RRC filtering for 80 Gbaud QPSK signal makes the constellation points appear more concentrated. This is the reason why the constellation diagram at 80 Gbaud looks better than that at 60 Gbaud in the original manuscript.

In the revised manuscript, we use the same digital signal processing (RRC filters) for both 60 Gbaud and 80 Gbaud, and we update the results in Fig.5 in the revised manuscript. The updated results of Fig.5b is as follows:

For the 16-QAM measurement, the device is over driven from the constellation diagram. Lower driving voltage is suggested to exploit the performance of the device, especially for Fig. 5(h).

Response: Thank you very much indeed for your suggestion. We have retested the 60 Gbaud 16QAM modulation with lower driving voltage and optimized filtering in the digital signal processing.

Accordingly, we have revised the Fig. 5h and Fig.5f for the revised manuscript:

Reviewer #2 (Remarks to the Author):

This paper presents what is claimed to be the first demonstration of an IQ modulator in thin film lithium niobate. The structure is traditional, consisting of two parallel Mach-Zehnder interferometers nested within a further Mach-Zehnder structure with a 90 degree phase shift introduced before combination.

The Mach-Zehnder structures are very traditional similar to [1] with velocity and impedance matched electrodes in a push-pull configuration. The device length in this submission are 7.5mm and 13mm. The Loncar demonstration [1] has longer electrodes (10mm and 20mm), but with similar geometry. Loncar shows a V_{π} of 1.4V with a bandwidth over 40GHz, while this submission demonstrates 1.9V with a bandwidth over 60GHz. So the drive voltage is not the highest reported.

The bandwidth is also not the highest reported. The publication [2] by Prather shows bandwidths can exceed 100GHz (approaching 1 THz) - however with a degraded V_{π} of 3.8V.

Response: Thank you very much indeed for your very professional comment. We have fabricated a new batch of devices designed to achieve record low half-wave voltages while maintaining a high modulation bandwidth. The arm lengths of newly fabricated LNOI IQ modulator is 18 mm. The measured V_{π} for the device is 1.25 V, corresponding to $V_{\pi}L$ of 2.25 Vcm, and the measured EO bandwidth is greater than 43 GHz. This performance metric is by far the best of its kind. We have updated the device information in the revised manuscript (**Supplementary Note 6| Modulator energy considerations**):

“We can further reduce the energy consumption of the LNOI-based IQ modulator through reducing the V_{π} . We fabricated a MZM with a long modulation length of 18 mm. In Supplementary Figure 7, it demonstrates a record low V_{π} of 1.25 V and high EO bandwidth of 43 GHz.”

Fig. S5 **a** Normalised optical transmission as a function of the applied voltage, showing V_{π} of 1.25 V. **b** Small signal response EO bandwidths (S_{21} parameter) of the 18-mm Mach-Zehnder modulator.”

The submitted paper quotes an optical insertion loss of 0.5dB/cm, while [1] quotes 0.5dB for the entire structure. The submitted paper does not report on the total insertion loss of the chip and in particular does not report the insertion loss due to propagation in proximity of the modulation or bias electrodes - it is important that these numbers be quoted as it is important that the electrodes do not interact strongly with the light for practical operation.

Response: Thank you very much indeed for your comment. The propagation loss of our LN

waveguide was measured to be 0.3 dB/cm for single mode waveguide and 0.15 dB/cm for 4- μ m-wide waveguide. The data was obtained by measuring the Q factor of the micro-ring resonator co-fabricated on the same chip with the modulator devices. The details of how we measure the propagating loss in the revised supplementary information.

We carefully designed the gap between the waveguide and the metal electrode. The calculated metal induced loss is around 0.04 dB/cm, which is negligible. In order to make sure that the modulation and bias electrodes does not indeed incur any unwanted insertion loss, we fabricated micro-ring resonator with and without metal electrode. For the micro-ring resonator with metal electrode, the gap between the waveguide and the electrode are designed to the same value as that in the modulator device. The measured Q factors are almost identical for both cases, which indicates that the metal electrode dose not incur any absorption loss. We include the details in the **Supplementary Note 3 and Supplementary Figure 4**.

There are two kinds of optical waveguides in our devices: 1- μ m-wide single mode waveguide and 4- μ m-wide multimode waveguide. To investigate the propagation loss of these waveguide, we fabricated LN microrings with the 1- μ m-wide waveguide and 4- μ m-wide waveguide. The etching depth of the microring waveguide are the same as the LN waveguide used in our modulators. From the measured Q factors of the transmission spectra (Supplementary Figure 4a,c), we can obtain the propagation loss is 0.3 dB/cm (1- μ m-wide waveguide) and 0.15 dB/cm (4- μ m-wide waveguide), respectively.

To further confirm the additional loss from bias and modulation electrodes, we fabricated Ti electrodes along the microring with 1- μ m-wide single mode waveguide and Au electrode along the microring with 4- μ m-wide multimode waveguide, respectively. The gaps between the electrodes and the waveguides were the same as those used in the modulator device. The Q factors were measured again for the microrings with electrodes (Supplementary Figure 4b,d), and the loss measured loss results indicates that the electrodes do not incur substantially additional absorption loss.

Supplementary Figure 4| Optical transmission spectra of the microring resonators. Measured transmission spectra (blue dots) of and Lorentz fittings (red lines) of a 1- μ m-wide LN microring resonator, b 1- μ m-wide LN microring resonator with Ti electrodes, c 4- μ m-wide LN racetrack resonator and d 4- μ m-wide LN racetrack resonator with Au electrodes.

We also give the detailed loss metric for the devices:

In the **Maintext**: “The fibre-to-fibre insertion losses at peak transmission are measured to be 8.6 dB and 8.25 dB for 13mm and 7.5mm device, respectively. The coupling loss of the grating couplers is 3.4 dB/facet. Therefore, the on-chip losses are 1.8 dB and 1.45 dB for 13mm and 7.5mm device, respectively. The coupling loss can be further improved by replacing grating coupler with edge-coupled spot-size converters, with fibre-to-fibre losses below 4 dB practically achievable.”

The submitted manuscript devotes a significant portion of the paper to reporting on the thermo-optic phase controllers. After a quick scan of the literature, I did not find any other report of thermo-optic switches on LNOI, so this may be innovative. However, the analysis done by the authors is not convincing. While electro-optic behaviour can be assessed based on the voltage required, the thermo-optic switches cannot be compared on voltage alone. The current must also be considered in order to assess the power draw.

Response: Thank you very much indeed for your comment. The Figure 2.a is changed into a plot of transmission against power consumption of the TO heater

I also do not agree with the authors about the 'LN is well-known for its drifts in the DC bias point upon the application of a static electric field, which is a phenomenon that originates from the piezoelectric nature of the material'. The authors must provide a reference for this statement. Bias control is necessary for interferometric lithium niobate devices due to the pyroelectric effect (changes in temperature cause a change in surface charge which in turn changes the refractive index). I know of many lithium niobate devices that do utilise an electro-optic bias and are able to maintain stability. Perhaps the authors are effectively controlling the temperature of the LNOI film with their TO heaters? I do not believe that their analysis of the instability of their own device when driven electro-optically provides any scientific insight and hence, I would encourage presenting the thermo-optic switches in a more technical journal where it can be the focus and referencing this in a higher profile paper describing the 'world first' IQ measurement.

Response: Thank you very much indeed for your insightful comment. The DC drift phenomenon originates from both extrinsic and intrinsic sources. The extrinsic origins of the drift are linked to the change of the effective refractive index of the LN via the possible variations of the temperature of the device (thermo-optic and pyroelectric effects), of the optical power in the waveguide (photorefractive effect) and the strain relaxation at the silica buffer-substrate interface (strain-optic effect). The intrinsic origin of the drift is related to the flow and redistribution of electrical charge under the application of the voltage. The electrical charges accumulate at z-cut surfaces which is the surface vertical to the self-polarization. While the extrinsic origins of the drift can be suppressed in principle, the intrinsic origin of the drift can only be mitigated, instead of eliminated. This DC drift effect is an intensively studied topic in conventional LN modulators, and companies manufacturing

commercial LN modulators have been working hard to mitigate or suppress the DC drift effect. More details can be found in: [IEEE Photon. Technol. Lett. 16, 2460-2462 (2004); J. Lightwave Technol. 29, 1522-1534 (2011); Broadband Optical Modulators, CRC Press 2011 Chapter 14 and 15].

Figure R1. (a) The exposed z-cut surface is the source for pyroelectric charges. (b)-(c) Applied sawtooth electrical signal for 10 Hz and 10 KHz and the measured modulation response of a 9-mm-long device.

In our case, the z-cut surfaces are exposed by dry etching process and the distance between the exposed z-cut surface and the electrode is very close, as depicted in Fig. R1(a). According to ref. [Broadband Optical Modulators, CRC Press 2011, page 350], the drift effect tends to become larger in modulator designs with larger exposed area of z-cut surface or closer distance between exposed z-cut surface and a center electrode. Therefore, the DC drift effect tends to be stronger than that in conventional LN modulators.

To study the drift effect in our device, we applied sawtooth modulation voltages of various frequencies to the 13-mm long modulator, and Fig. R1(b) and Fig. R1(c) shows the measured results for 10 Hz and 10 KHz, respectively. For a slowly-changed applied voltage, that is 10 Hz case, the response of the modulator shows fluctuations when the voltage changes or jumps abruptly (see the part identified by the dashed circle). We attribute this phenomenon to DC drift in LiNbO₃ as described above. For a fast-changed applied voltage, that is 10 KHz case, the fluctuations in the response curve disappear and influence of the DC drifts is suppressed. These results, together with the results we put in the manuscript (Fig.2 b), confirms that DC drift effect is there when applying a DC voltage or a slowly changed voltage.

We would like thank you very much for your suggestions about writing a detailed paper about the thermal optic switches, and we are doing that exactly.

In the revised manuscript, we add three reference for the DC drift in LN modulators:

11. Chen, A. & Murphy, E. *Broadband Optical Modulators*. (CRC Press, City 2011).
48. Nagata, H., Brien, N. F. O., Bosenberg, W. R., Reiff, G. L. & Voisine, K. R. DC-Voltage-induced

thermal shift of bias point in LiNbO₃ optical Modulators. *IEEE Photon. Technol. Lett.* **16**, 2460-2462 (2004).

49. Salvestrini, J. P., Guilbert, L., Fontana, M., Abarkan, M. & Gille, S. Analysis and Control of the DC Drift in LiNbO₃-Based Mach-Zehnder Modulators. *J. Lightwave Technol.* **29**, 1522-1534 (2011).

Overall, the LNOI technology presented here does not, in my view, represent any significant advancement in the state of the art; however, the implementation of a IQ modulator may indeed be a world first, and perhaps this is significant enough to warrant publication in a high profile journal such as Nature Communications. However, in order to achieve this level, I think the focus needs to be placed more on the system demonstration that harnesses the IQ modulator (rather than the technology used to realise the chip itself) and it would need to be demonstrated that a record breaking transmission characteristic was achieved (such as modulation density, or power efficiency?).

Response: Thank you very much indeed for your insightful comment. We have added the following work in the revised manuscript in order to increase the impact of our paper.

1. We fabricated a new batch of devices with a longer modulation length (18mm) to achieve record low half-wave voltages while maintaining a high modulation bandwidth. The V_{π} is 1.25V and the EO bandwidth is 43GHz, which surpass the value reported in Loncar's group. Please find the details in the above text.

2. In order to better compare TO phase shifter and EO phase shifter, we fabricated an LNOI-based IQ modulator with EO phase shifters for DC bias voltage control. The TO and EO phase shifter has both advantages and disadvantages. The TO phase shifter generates extra power consumption compared to the EO phase shifter, but the size is much compact and the bias control is more stabilized. The details of IQ modulator with EO phase shifters are added in the revised supplementary information:

In the Supplementary Note 5 | IQ modulator with EO phase shifters: *“To further compare the devices with TO phase shifter and EO phase shifter, we fabricated an LNOI-based IQ modulator with EO phase shifters for DC bias voltage control (Supplementary Figure 6a). In Supplementary Figure 6b, the 3-mm-long EO phase shifter was used to introduce a 90° phase difference between the In-phase/Quadrature branches. It takes about 5V to set the modulator to the optimum modulation bias point. We use EO phase shifters to adjust the IQ modulator operating at the optimum bias points to generate 64 Gbaud QPSK signals. We can observe that constellation diagram of the QPSK signals became seriously degrade and the BERs deteriorate from 8.6×10^{-5} to 7.4×10^{-3} in 5 minutes (Supplementary Figure 6 c-d). It turns out that the EO effect can be used to control the LNOI modulator but we need to use bias control board to maintain a stable operation.*

Supplementary Table 1 compares several performance metrics of the demonstrated IQ modulators with TO and EO phase shifters. The modulators show no difference in the value of V_{π} and EO bandwidth because of the same travelling-wave electrodes and optical waveguides design. The TO phase shifter generates extra power consumption compared to the EO phase shifter, but the size is much compact and the bias control is more stabilized.

Supplementary Figure 6 | ***IQ modulator with EO phase shifters.*** **a** Schematic of an integrated LN IQ modulator with EO phase shifters for DC bias voltage control. **b** Microscope image of a fabricated device. **c, d** Measured constellation diagrams for 64 Gbaud 16 QAM signals under optimum bias points **c** at once and **d** 5 minutes later.

Supplementary Table 1. Comparison of several performance metrics of the 7.5-mm IQ modulators with TO and EO phase shifters

	V_{π} (V)	3-dB EO bandwidth (GHz)	Length of phase shifter (mm)	Power consumption caused by phase shifters (mW)	Whether the bias control is stable
IQ modulator with TO phase shifters	3.1	> 67	0.16	~16.6	Yes
IQ modulator with EO phase shifters	3.1	> 67	3	0	No

3. We calculate the energy consumption for our 16 QAM modulation. Accordingly, we have added the following texts for the revised manuscript:

In the **Methods**: “**Modulator energy considerations.** For 16 QAM modulation, we can estimate the energy consumption per bit dissipated in the travelling-wave IQ modulator as $W_{bit,16QAM} = 2 \times V_{rms}^2 / (BR)$, where V_{rms} is the root-mean-square voltage of the electrical PAM4 signal, B is the total bit rate and R is the equivalent resistor of 50Ω . The value of V_{rms} applied to the one MZM was measured directly using the oscilloscope. For 60 Gbaud 16 QAM ($B = 240 \text{ Gbit s}^{-1}$) modulation experiment with $V_{rms} = 0.85 \text{ V}$, we can calculate an energy consumption of 120 fJ/bit. For 80 Gbaud 16 QAM ($B = 320 \text{ Gbit s}^{-1}$) modulation experiment with $V_{rms} = 0.7 \text{ V}$, we find a lower power consumption of 61 fJ/bit. To further reduce the energy consumption, we fabricated a LN MZM with a record low V_{π} of 1.25 V while maintaining a high modulation bandwidth (see Supplementary Note 6 and Supplementary Fig. 7).”

I also note a number of typographical errors throughout the manuscript (for example "both branches of the 13-mm and 7.5-mm devices as a function of the applied voltage, showing V_{π} of 1.9 and 7.5 mm, respectively." - the last number should be the V_{π} of the 7.5mm electrode, not its length). There are several others.

Thank you very much indeed, and sorry for the careless mistakes. We have thoroughly proofread our manuscript.

[1] Wang, C., Zhang, M., Chen, X. et al. Integrated lithium niobate electro-optic modulators operating at CMOS-compatible voltages. *Nature* 562, 101–104 (2018)

[2] Mercante AJ, Shi S, Yao P, Xie L, Weikle RM, Prather DW. Thin film lithium niobate electro-optic modulator with terahertz operating bandwidth. *Optics express*. 2018 May 28;26(11):14810-6.

Reviewer #3 (Remarks to the Author):

Lithium niobate is one of the most important materials for optical modulators. With the LNOI (lithium niobate on insulator) platform, new types of optical modulators were introduced in recent years. This kind of modulators has the advantages of small volume and high performance, such as low driving power, high bandwidth, etc. However, LNOI based modulators still face several important questions.

1. Can this modulator be used on short reach links, such as metro and data-center interconnects? Modulators for short reach links account for the most amount of the modulator industry production. In such application, the basic requirements for the modulator are small volume, low optical loss, low power consumption, and low cost. Currently the standard choice is the semiconductor (InP, for example) based modulators. InP modulator has the drawbacks of high optical loss and non-linearity. LNOI waveguide showed an ultra-low loss down to 0.03 dB/cm (*Optica* 4, 1536, 2017), and conventional LN modulator showed a good linearity. Therefore, LNOI modulator has a potential to be a strong competitor with InP modulators.

2. For LNOI modulator, which fabrication routine will be adopted? In other words, what is the future technology roadmap? Two complementary fabrication routine were used to make LNOI based modulators: monolithic and hybrid. Monolithic means direct etching of LN to form the waveguides (*Nature* 562, 101, 2018). This kind of modulator theoretically has a better device performance. An example of hybrid is SOI-LNOI modulator (*Nat. Photon.* 13, 359, 2019). This modulator has the advantage of the mature fabrication technology, and easiness to integrate with photo detectors (such as epitaxy of SiGe detector). Which fabrication routine will be the mainstream? This question is still not clear to the academy and the industry. The people in industry especially care about this question, because they concern the R&D inputs. 3. For conventional LN modulator, a major problem is DC drift. This phenomenon is attributed to the piezoelectric or pyroelectric properties of the LN material. People spent a lot of efforts to solve this problem. For LNOI modulator, because it is a new research field, there is very few data or public report on the DC drift problem.

This manuscript reported the first monolithic IQ (in-phase/quadrature) LNOI modulator used for short reach links, solved the DC drift problem using TO (thermal-optic) phase shifters. The device dimension is small (15 μm), and the performance is excellent, such as low loss

(1.45 dB), low $V\pi L$ (2.5 V cm), high data rate (320 Gbit s⁻¹), and large bandwidth (> 67 GHz).

The drawback of this work is that TO phase shifters needed electric power to operate, which increased the power consumption.

Response: Thank you very much indeed for the Referee's very professional comment. We totally agree. The advantages of the TO phase shifters are shorter device length and more stable DC bias, but the drawback is that it consumes power and it will age when subjected to extreme temperatures. We believe TO phase shifter and EO phase shifter both have their own advantages and disadvantages. They can be useful in different application scenarios. For example, the TO shifter can save the device length for several millimeters, which is a great if people want to operate the IQ modulator in a very compact module, say, QSFP-DD. The EO shifter could be useful in scenarios where the device length is not a priority. In order to clarify this, we revised the manuscript in following way:

1. The figure 2.a is changed into a plot of transmission against power consumption of the TO heater (the original figure 2.a is a plot of transmission against applied voltage).
2. We add one sentence in the manuscript to compare the pros and cons of TO phase shifter and EO phase shifter.
3. We fabricated new IQ modulators with EO phase shifter and we put the details of the device and measurement results in the **Supplementary Note 5** to provide a better comparison. We also added the sentences in the maintext to clarify this information.

In the **Maintext:** "It should be noted that the TO phase shifter consumes static power, while EO phase shifter does not. We also fabricated an IQ modulator with EO phase shifters for further comparison (Supplementary Note 5, Supplementary Fig. 6 and Supplementary Table 1)."

This work provided the strong evidences to answer the three questions mentioned above. It will be very interesting to the people working on the modulators.

A publication of the manuscript is highly recommended. However, the authors should response the following comments before the publication.

1. The authors can give some comments on the comparison on the modulators fabricated by hybrid or monolithic. Since the authors had the experience and insight on SOI-LNOI hybrid modulator (Nat. Photon. 13, 359, 2019), and this manuscript was about monolithic modulator, the authors' comments would be very useful to the readers.

Response: Thank you very much indeed for your suggestions.

The monolithic LNOI device can be regard as the next generation of the widely used conventional lithium niobate modulators. The performance of the LNOI devices is much better than the conventional counterpart, and more importantly, the footprint of the LNOI can fulfil the requirement of certain emerging application scenarios. Moreover, the manufacture of LNOI devices does not involve any complicated processes. Therefore, we believe LNOI device can find practical applications in the near future. In contrast, SOI-LNOI hybrid device is a technology for longer term. Silicon photonics are promised to create a radically new landscape for photonic integrated circuits. By harnessing the tool sets and process flows in CMOS foundries, the technology offers advantages of low-cost, high-volume and reliable manufacturing. However, the performance of silicon optical

modulator is limited by the lossy and nonlinear carrier effect. This is where the SOI-LNOI hybrid modulators come in. By combining LNOI with silicon photonics, the hybrid platform allows for the combination of 'best-in-breed' active and passive components, offering the solutions for high-performance silicon based modulator. Therefore, the SOI-LNOI hybrid device is aimed for improving the silicon photonic platform. In the future, SOI-LNOI hybrid device has to find its way to the compatibility of CMOS process in the foundry before the its practical applications. In the future, this hybrid technology can combine 'best-in-breed' components, including thin-film lithium niobate modulators, III–V semiconductor, and silicon circuits, which is very important to address the cost and energy crunch.

To clarify this, we have added the following texts in the supplementary information for the revised manuscript:

Supplementary Note 1: *"The monolithic LNOI device can be regard as the next generation of the widely used conventional lithium niobate modulators, which can find practical applications in near future. In contrast, SOI-LNOI hybrid device is a technology for longer term. Silicon photonics is becoming the leading technology for photonic integrated circuits. By harnessing the tool sets and process flows in CMOS foundries, the technology offers advantages of low-cost, high-volume and reliable manufacturing. Recently, thin-film LN is co-integrated on silicon photonics in a hybrid way, which opens up new avenues for high-performance silicon based optical modulators. This hybrid integration is promised to offer one solution for future photonic integrated circuits with high-level of integration, which combines the 'best-in-breed' components, including thin-film LN modulators, silicon passive devices and III-V active devices. "*

2. The authors can give some comments or data on the multimode interference couplers (MMIs) in this work. There is few report about MMI coupler on LNOI. Any information about it will be useful.

Response: Thanks very much for your suggestion. We have added the following texts and figures in the **Supplementary Note 2 and Supplementary Fig. 1** for the revised manuscript about LN MMI coupler:

"We employ 1×2 multimode interference (MMI) couplers to provide symmetric power splitting and combing in our modulators. The MMI couplers considered here are based on the etched LN waveguides with a slab thickness of 300 nm and a rib height of 300 nm. The length of the multimode region L_M was designed to be $64 \mu\text{m}$, and the width of the multimode region W_M and the wider ends were set as $9.654 \mu\text{m}$ and $4.54 \mu\text{m}$, respectively (Supplementary Figure 1a). Three identical $50\text{-}\mu\text{m}$ -long linear tapers are used as adiabatic tapers and connected to single-mode waveguides. Supplementary Figure 1b shows the simulated excess loss for different wavelengths.

To measure the excess loss of MMI coupler, we fabricated the MMI couplers arranged in a cascaded configuration, as depicted in Supplementary Figure 1c. Supplementary Figure 1d shows the measured transmission of the MMI couplers at the wavelength of 1550 nm. We extract the excess loss of $\sim 0.054 \text{ dB}$ from the slope of the fitting line. Next, we experimentally demonstrated the performance of our 3-dB MMI couplers by integrating two MMI couplers into an unbalanced MZI (Supplementary Figure 1e). Supplementary Figure 1f shows the static extinction ratio is

greater than 40 dB, which indicates that the power splitting ratio of the MMI is very close to 50:50.”

Supplementary Figure 1 | MMI performance. *a* Schematic of an MMI coupler. *b* Simulated excess loss of the MMI for different wavelengths. *c* Schematic of the MMI couplers arranged in a cascaded configuration for loss measurement. *d* Transmittance measured at the output of the cascaded MMI couplers as a function of the number of branches (blue circle) a linear fit (red solid line). *e* Schematic of an unbalanced MZI with two 1×2 MMI couplers. *f* Measured static extinction ratio of the unbalanced MZI.”

3. The authors used TO phase shifter to suppress the DC drift, but the TO effect would increase the power consumption of the modulator. Can EO effect be used for DC drift control in LNOI modulator?

Response: Thank you for your question. The TO heater does increase the power consumption of the modulator. But low power consumption can also be realized in the TO heaters through removing substrate to provide thermal isolation. We can apply this technique to the LNOI platform.

Besides, we fabricated another chip with IQ modulator, and all DC biases were controlled by EO effects on the modulator. The bias voltages were applied to I and Q branches via bias-T, and the 3-mm-long EO phase shifter was used to introduce a $\pi/2$ phase difference between two sub-MZMs. It turns out that the EO effect can be used to control the LNOI modulator but we need to use bias control board to maintain a stable operation.

Accordingly, we have added the following texts and figures in the supplementary information for the revised manuscript:

In the **Supplementary Note 5 | IQ modulator with EO phase shifters:** “To further compare the devices with TO phase shifter and EO phase shifter, we fabricated an LNOI-based IQ modulator with EO phase shifters for DC bias voltage control (Supplementary Figure 6a). In Supplementary Figure 6b, the 3-mm-long EO phase shifter was used to introduce a 90° phase

difference between the In-phase/Quadrature branches. It takes about 5V to set the modulator to the optimum modulation bias point. We use EO phase shifters to adjust the IQ modulator operating at the optimum bias points to generate 64 Gbaud QPSK signals. We can observe that constellation diagram of the QPSK signals became seriously degrade and the BERs deteriorate from 8.6×10^{-5} to 7.4×10^{-3} in 5 minutes (Supplementary Figure 6 c-d). It turns out that the EO effect can be used to control the LNOI modulator but we need to use bias control board to maintain a stable operation.

Supplementary Table 1 compares several performance metrics of the demonstrated IQ modulators with TO and EO phase shifters. The modulators show no difference in the value of V_π and EO bandwidth because of the same travelling-wave electrodes and optical waveguides design. The TO phase shifter generates extra power consumption compared to the EO phase shifter, but the size is much compact and the bias control is more stabilized.

Supplementary Figure 6| IQ modulator with EO phase shifters. **a** Schematic of an integrated LN IQ modulator with EO phase shifters for DC bias voltage control. **b** Microscope image of a fabricated device. **c, d** Measured constellation diagrams for 64 Gbaud 16 QAM signals under optimum bias points **c** at once and **d** 5 minutes later.

Supplementary Table 1. Comparison of several performance metrics of the 7.5-mm IQ modulators with TO and EO phase shifters

	V_π (V)	3-dB EO bandwidth (GHz)	Length of phase shifter (mm)	Power consumption caused by phase shifters (mW)	Whether the bias control is stable
IQ modulator with TO phase shifters	3.1	> 67	0.16	~16.6	Yes
IQ modulator with EO phase shifters	3.1	> 67	3	0	No

4. The authors have to check the content of the manuscript carefully to avoid any

miswriting, for example, in line 73, “Fig. 2b” should be “Fig. 1b”.

Response: Thank you very much indeed, and really sorry for the careless mistakes. We have thoroughly proofread our manuscript.

REVIEWERS' COMMENTS:

Reviewer #2 (Remarks to the Author):

Many thanks to the authors for their thorough response to the questions raised in my original review. I believe that the manuscript is now significantly improved and I would suggest acceptance for publication.

Reviewer #3 (Remarks to the Author):

The authors had responded my comments carefully and satisfactorily. The manuscript can be published.